# Evaluation of Age Based-Sleep Quality and Fitness in Adolescent Female Handball Players

**DOI:** 10.3390/ijerph20010330

**Published:** 2022-12-26

**Authors:** Mohamed Alaeddine Guembri, Ghazi Racil, Mohamed-Ali Dhouibi, Jeremy Coquart, Nizar Souissi

**Affiliations:** 1Research Unit: Physical Activity, Sport and Health (UR18JS01), National Observatory of Sports, Tunis 1003, Tunisia; 2Research Unit 17JS01 (Sport Performance, Health & Society), Higher Institute of Sport and Physical Education of Ksar Saîd, Manouba, Tunis 2010, Tunisia; 3Laboratory of Clinical Psychology: Intersubjectivity and Culture, Faculty of Humanities and Social Sciences, University of Tunis, Tunis 1007, Tunisia; 4Univ. Lille, Univ. Artois, Univ. Littoral Côte d’Opale, ULR 7369-URePSSS-Unité de Recherche Pluridisciplinaire Sport Santé Société, Lille, France

**Keywords:** sleep quality, puberty, sleep disorders, physical condition, insomnia, sleepiness, body fat, lean body mass

## Abstract

The present study aimed to examine the differences in sleep hygiene, balance, strength, agility, and maximum aerobic speed (MAS) between two groups of female handball players aged under 14 (U14) and under 17 (U17) years. Seventy-two female handball players participated and were divided into two groups according to age: U14 (*n* = 36, age: 13.44 ± 0.5 years) and U17 (*n* = 36, age: 15.95 ± 0.76 years). Sleep hygiene was evaluated using three questionnaires: Sleep quality and sleepiness via the Pittsburgh (PSQI) and Epworth (ESS) questionnaires, and the insomnia questionnaire via the measurement of the insomnia severity index (ISI). Physical fitness was evaluated with the stork balance tests with eyes open (OEB) and closed (CEB), the vertical jump (SJ), horizontal jump (SBJ), and five jump (FJT) tests, the agility (*t*-test) and the maximum aerobic speed (MAS) tests. No significant differences were shown between U14 and U17 players in all PSQI, ISI, and ESS scores, and balance and strength performances. Meanwhile, the U17 players’ performances were significant better in agility quality (*p* = 0.003 < 0.01) and MAS (*p* = 0.05) compared to the U14 players. Biological gender specificity during the maturation phase may inhibit the improvement of balance, and strength performances between the age of 13 and 17 years, while agility and MAS performances are more affected by age alterations.

## 1. Introduction

Handball is a team sport with an intermittent nature involving several physical qualities such as balance [1,2], muscular strength [3,4], agility [5,6], and endurance [7,8]. Thus, improving these qualities is essential for development during childhood and adolescence [9].

Previous studies have shown the importance of this evaluation to distinguish the elite players from the sub-elite players of the same age category [10,11]. It was demonstrated, however, that the reactive agility quality presented, for example, a distinguishing factor for experienced versus amateur players [12]. Although basic motor abilities have been evaluated in women’s handball [13], to the best of the authors’ knowledge, no studies have examined the difference that may occur in some measures of physical condition in the young players.

Mostly, young athletes’ training sessions are based on technical and tactical work and less on the development of physical abilities, which change with age and are affected by biological maturation [14]. It is important to mention that the stabilization of fitness is noted between the beginning and end of puberty [15], and that girls’ morphological development is marked by a high intensity of fat mass accumulation at that phase, which represents an inhibiting factor for performance [16].

Understanding the effects of changes in girls during the transition from pre-puberty to post-puberty seems to be of interest for coaches to better plan and intervene, especially on the physical level, to optimize performance. Therefore, one of the objectives of our study was to focus on the differences in some measures of physical fitness in female handball players between the ages of under 14 and under 17. We hypothesized that performance is better for U17 compared to U14 for both anaerobic and aerobic tests. However, having 3 years’ experience in practice, may present a good argument to distinguish this difference during maturation regardless of gender.

Indeed, having sleep disorders such as insomnia can inhibit the proper functioning of cognitive mechanisms such as attention, concentration, and memory, which are factors in sports performance. These troubles, which were suggested to be caused by fatigue and daytime sleepiness [17], are related to poor physical condition [18], including cardiorespiratory capacity [19]. This prompts us to focus on the evaluation of certain aspects of sleep in young athletes in their adolescent phase.

This period of adolescence undergoes dramatic changes in sleep [20,21]. In this context, two processes intervene in its regulation. The first one targets the intrinsic circadian system, and the second one targets the homeostatic sleep–wake system [22]. This context, however, explains the short sleep duration, during either the nights and even the day of sports training [23,24].

It was reported that many adolescents suffer from the prevalence of sleep disorders such as sleepiness [17] and insomnia [25,26]. As recently shown [27], more than 39% of adolescents suffered from poor sleep quality. In this regard, it is crucial to develop comprehensive sleep education in adolescents at first, and which should be respected in the school curriculum.

Furthermore, this finding leads us to evaluate the nature of sleep hygiene by establishing a comparison of the quality of sleep via the Pittsburgh questionnaire (PSQI), the insomnia questionnaire via the measurement of the insomnia severity index (ISI), and sleepiness via the Epworth questionnaire (ESS), between U14 and U17 female players.

However, the objectives of the present study were (1) to compare sleep parameters (PSQI, ISI, and ESS) between U14 and U17 female players and (2) to compare the differences in some fitness measures between these two different age groups.

## 2. Materials and Methods

### 2.1. Participants

Seventy-two healthy female handball players participated in this study. They were divided into two different age groups: U14 (36 players, age: 13.44 ± 0.5 years, height: 1.64 ± 0.04 m, BM: 57.83 ± 5.79 kg, BMI: 21.49 ± 1.52 kg m^−2^) and U17 (36 players, age: 15.95 ± 0.76 years, height: 1.67 ± 0.05 m, BM: 59.21 ± 6.16 kg, BMI: 21.22 ± 1.63 kg m^−2^). All of them belong to a regional local handball team, and they train for 1 h 30 min per session 4 times weekly. Before beginning the intervention, all participants and their parents signed an informed consent form in accordance with the international ethical standards, in particular the Declaration of Helsinki [28]. It should be noted that four participants withdrew before the start of the study for personal reasons (2 from each group) and their data were not accounted for in the statistical analysis.

### 2.2. Anthropometric Measurements

All anthropometric measurements were performed in the afternoon at the end of the week with the help of a specialized physician.

Body height was measured in centimeters with no shoes, heels together, and the back of the subject parallel to the stadiometer (Model 214 height rod; Seca, Hamburg, Germany). Body mass (BM) was assessed to the nearest 0.1 kg with a digital scale (Tanita, Tokyo, Japan) and the body mass index (BMI = Mass [kg]/(Height [m])^2^) was determined. The fat mass (FM), lean mass (LM), and body mass index (BMI) were measured for each participant by bioelectrical impedance analysis (BIA) (Tanita Body Composition Analyzer Mode TBF-300, Tokyo, Japan).

With the help of a qualified pediatrician, assessment of the pubertal stage was determined according to the Tanner classification [29] (Tanner and Whitehouse 1976): Pubertal children included Tanner stages II-III and post-pubertal children were in Tanner stages IV-V (refer to Table 1).

### 2.3. Procedure

The experimental part of this study was spread over a vacation period of 3 days (Figure 1). Before the commencement of the experimentation, a physician made sure that the players were not sick, did not take any medication, and had not practiced any sport on that day. Thus, all tests were performed in a single session starting at 4:00 pm since performance peaks in the late afternoon for anaerobic tests and in accordance with the hours of the day in which most of the training sessions were regularly performed as determined by Chtourou et al [30], except for the VAMEVAL test, which was performed in the second session.

Day1:

On the first day, the 2 groups completed the three sleep questionnaires. Thus, each participant had to answer questions about their sleep attitudes during the days and nights of the last month: The Pittsburgh Questionnaire (PSQI), the Insomnia Questionnaire (ISI), and the Sleepiness Questionnaire (ESS). Each of these is composed of several items. To answer the questionnaires, an explanation was presented by an examiner who gave a verbal signal to the participant who passed from one item to another.

The Pittsburgh sleep quality index (PSQI) [31] is a questionnaire for the subjective evaluation of sleep quality; it is composed of 19 questions combined into 7 scores. The 7 component scores are added together to obtain an overall score ranging from 0 to 21, and an increase in the score coincides with a decrease in sleep quality.

Moreover, the insomnia severity index (ISI) [32] is a self-reported subjective measure of insomnia symptoms and the levels of worry caused by sleep disorders, composed of seven items. When adding up the scores, it helps to give an overall score ranging from 0 to 28. Therefore, the scores between 0 and 7 = No insomnia; 8–14 = Subclinical insomnia (mild); 15–21 = Clinical insomnia (moderate); and 22–28 = Clinical insomnia (severe).

On the other side, the Epworth Sleepiness Scale (ESS) [33] is a self-administered questionnaire; it is composed of 8 items that measure the “usual probability of dozing or falling asleep” in common everyday situations. The ESS score ranges from 0 to 24, and when the score is higher than 10 it is an indicator of severe drowsiness.

Day2:

The second day of the intervention was dedicated to the realization of the tests in the indoor handball court. Each training session started with a warm-up of approximately 10-min based on running, joint mobilization, and stretching followed by five bursts of 20-m sprints [34].

Participants also performed the balance test “stork balance” with eyes open and eyes closed. This test consists of maintaining balance on one leg and on the sole of the foot for as long as possible.

Except for the test of the squat jump (SJ), which was performed in door a Handball court, the standing long jump (SBJ) and the five-jump tests (FJT) were performed outdoors on an athletic field. In the agility test “*t*-Test”, the participant accelerated and decelerated with rapid changes of direction. When the players had completed all repetitions, they moved from one test to the other. The passive recovery time was fixed to 5-min. It is important to note that a familiarization test was performed at the beginning of the intervention to eliminate any learning factor that could bias the results. Four investigators assured the completion of all the tests. All participants were encouraged verbally. However, all measures were again collected and performed (in the same conditions) for comparison with baseline values.

#### 2.3.1. Stork Balance Test

This test was performed as described by Sopa [35]. The participant was required to stand on a single leg of their choice, barefoot and on a flat surface. The other leg was raised so that the foot of that leg was glued next to the knee of the supporting leg. The participant had to put her hands on her hips. The clock started when the participant lifted the heel of the supporting leg. The participant had to maintain this position for as long as possible, and the timer stopped when she dropped the heel of the supporting leg, rotated the body in any direction, or lifted the hands from the hips. Each player had 3 attempts to practice balancing either with eyes open (OEB) or closed (CEB). The participant was allowed to practice their balance for one minute and the total time was recorded. The best result of the three attempts was taken into consideration.

#### 2.3.2. Squat Jump (SJ)

This test involved evaluating the quality of jumps using an Opto-jump system [36]. Three trials were performed, and the best performance expressed in centimeters was retained for statistical analysis.

#### 2.3.3. The Standing Broad Jump or the Horizontal Jump (SBJ)

This test was performed as indicated by [37].

From the standing position, both feet had to touch the starting line and the player had to jump as far as possible in a horizontal direction. The take-off and landing phases of the jump had to be performed with both feet. The distance from take-off to the heel of the nearest foot on landing was measured in centimeters. Three trials were performed, and the best performance was taken for later analysis.

#### 2.3.4. Five Jump Test (FJT)

The participant performed five successive horizontal jumps [38], which started with their feet together and with an upright posture. The participant executed five bouncing strides on one leg, elevating the free knee and the opposite arm towards the front, and finished with both legs together. The distance covered was measured and expressed in meters. Three attempts were performed, and the best performance was taken into account.

#### 2.3.5. The Agility *t* Test

This test was administered as described by [39]. At the signal, the participant sprinted forward 9.14 m, touched the end of the first cone with her right hand, then ran laterally to the left for 4.57 m to touch the end of the second cone with the left hand and continue to move again for 9.14 m to the right and touch the end of the third cone with the right hand. The participant made a lateral return of 4.57 m to the left to touch the cone in the middle with the left hand. The test ended with a backward run to the starting point (9.14 m). A photoelectric cell (Photocells, Microgate^®^, Bolzano, Italy) was used to measure the performance. Three attempts were performed, and the best performance was taken into account.

#### 2.3.6. VAMEVAL Test

In order to assess the maximum aerobic speed (MAS), a running test was performed on a 200-m athletic track. This was administered as described by [34]. Blue cones were placed every 20-m at the lane line boundary of the track. Similarly, red cones were placed 2-m behind the blue cones. An examiner followed participants with a scoring table containing the beep times of the different levels, a stopwatch, and a whistle. The examiner made a short sound when the participant had to be next to the blue cone so that he could manage the running speed according to each level. At each whistle, the participant had to be within 2-m of the blue cones. When the participant did not follow the rhythm of the sound and was twice behind a red cone or when he stopped the race, the event was over. The maximum aerobic speed (MAS) corresponds to the level already finished. If the participant did not run the last stage for the full duration, then the method in [40] was used to calculate the MAS. The test started on the track at 8 km/h for 2 min, and for each stage of one minute, the speed increased by0.5 km/h.

Maximum aerobic speed in km/h was recorded. VO_2max_ was calculated using the formula proposed by [41,42]: VO_2max_ = 0.0324 × v^2^+ 2.143 × v + 14.49; where v is the speed of the last level expressed in km h^−1^ and VO_2max_ in mL kg^−1^ min^−1^.

### 2.4. Statistical Analysis

Analyses of all data were performed using SPSS version 26.0 (SPSS, Inc., Chicago, IL, USA). Results were presented as mean ± standard deviation. The categorization of the 3 sleep variables (PSQI, ISI, and ESS) was presented as percentages for the U14 and U17 athletes. For all the studied variables, the normality of the data distribution was checked using the Kolmogorov–Smirnov method.

A student’s *t*-test of independent samples was performed to compare the mean scores of PSQI, ISI, and ESS according to age (U14 and U17). Thereafter, the different measures of physical condition such as the average values of balance, strength, agility, and MAS between these 2 groups were also calculated. An alpha *p* value of less than 0.05 was used as a significance threshold.

## 3. Results

No significant differences were observed for the different sleep parameters (PSQI, ISI, and ESS) between the U14 and U17 players (Table 2).

The prevalence of players with poor sleep quality (PSQI ≥ 5) was 61.1% for the U14 group and 63.8% for the U17 group of older participants; mild insomnia (ISI ≥ 11) was 8.3% for the U14 group and 8.4% for the U17 group; sleep debt (ESS > 8) was 19.4% for the U14 group and 22.3% for the second group (see Table 3).

Although it was not significant, the balance performance with eyes open and closed for the U14 group was better (0.59 and 0.31, respectively).

In the between-group comparison, the agility values were higher (*p* < 0.01) in favor of the U14 group compared to the U17 group. Therefore, we reject the null hypothesis of equality of variances, and a significant difference between the two groups regarding agility performances was noted. Concerning the vertical (SJ) and horizontal (SBJ) jump tests, no significant difference was noted (*p* = 0.06 and *p* = 0.43, respectively).

Although the difference was not significant (0.06 > 0.05), the performance of the five-jump test (FJT) of the younger U14 players was better than that of the others, with a difference of 2% between the two groups.

On the other hand, the mean performance of the maximum aerobic speed capacity of the older U17 players was significantly higher compared to theU14 players (*p* < 0.05) (see Table 4).

## 4. Discussion

The current study examined the evaluation of the differences in PSQI, ISI, and ESS scores between U14 and U17 female players and of some measures of balance, strength, agility, and maximum aerobic speed.

Collectively, the main findings indicated, however, no significant differences in the different sleep parameters between the two groups of female players. However, significantly higher performances for the U17 category in agility quality and MAS, when compared to the U14 players, were noted.

Concerning the sleep parameters, this can be explained by the similarities in age between these two groups although they belong to different categories (U14: school A; U17: cadet B).

Furthermore, as the PSQI, ISI, and ESS scores increased in relation to the decreased sleep quality, this strengthens the idea of aptly targeting different participants’ age categories in order to have better performance. When using different age categories, Davenne [43] also found that sleep disorders such as nocturnal awakenings, short durations of deep sleep with slow waves, and problems with early awakening were highly prevalent with advancing age.

Furthermore, Rasekhi et al. [44], noted several determinants of the appearance of poor sleep quality and increased ISI in adolescents, i.e., gender, BMI, nature of sport, coffee consumption, daily activity, diet, and skipping breakfast. A recent study by Bruce et al. [45] even noted the importance of the biological factor at this age range, which causes a biological delay in the time taken to fall asleep. In fact, to attend school, most young are obligated to wake up too early, as is the fact for our female players. Hence, a sleep debt during the school week as shown in the current study, resulted in an in-crease in the ISI, ESS, and PSQI scores. However, sleep disorders, comprising poor sleep quality that exceeded 61% (PSQI ≥ 5), mild insomnia (ISI ≥ 11) which was approximately 8.3%, and sleep deprivation (ESS > 8) which was 19.4% for the U14 group against 22.3% for the U17 group, might all be considered as stimulating negative factors. These corroborate the study of Bel et al. [46] conducted on adolescents, which showed that the prevalence of sleep disorders is the result of the accumulation of several factors such as the relationship between insufficient sleep and poor nutrition. It is therefore a necessity to adopt educational strategies that may develop a culture of healthy lifestyles in order to have mentally and physically balanced adolescents.

Furthermore, Ferranti et al. [47] noted in one study that only 47% of adolescents followed the recommendations for daily sleep hours, while Carvalho [27] showed that more than 39% of adolescents had poor sleep quality and this was related to poor quality of life, such as nutrition or daily activities. According to Figueiro [48], a sleep phase characterized by long sleep latencies relative to desired bedtimes resulted in significant sleep deprivation in subjects’ daily activities.

From another perspective, we noted that the balance performance of the stork with eyes open and closed was better for the U14 group compared to the U17 group. We have to stress the fact that there were no previous studies having compared the unipodal balance quality of U17 and U14 female players in handball. Since unhealthy sleep habits increase with age, we suppose that this may be one of the factors that affected the behavior of the female players in the present study. In fact, the prevalence of sleep disorders has further been shown to affect the muscle strength of the lower limbs [49,50,51], which helped to maintain control of the posture through sensorimotor strategies. Maintained over a long period of time, this sleep disorder can deteriorate postural stability by acting on cognitive and biomechanical mechanisms [52,53,54,55]. According to Cerrah et al. [56], this requires may be the application of functional balance training for adolescents three times a week to improve static balance performance.

Concerning the participants’ average BMI values, the results showed that in the U14 group, these were significantly elevated compared to the U17 group. In fact, Rusek et al. [57] focused on sedentary adolescents of both sexes and showed that the increase in BMI values corresponded to better balance performance. In the following study, the 3 years of experience in sports practice for the U17 group were supposed to be sufficient in showing better sensorimotor performances resulting in a better balance and good postural control. Those results seem to be in disagreement with what was reported by Caballero et al. [58], showing that the level of expertise related to handball practice was a factor that improved postural balance. It is important to mention that his study was based on male handball players, and it may be that the increased testosterone secretion affected the noted anthropometric changes.

Regarding the between-group jump comparison (SJ, SBJ, FJT), they were indicative of the quality of the explosive strength using the lower limbs, even though no significant differences were noted. This study showed however that no significant difference was obtained for the comparison of the different jump performances (SJ, SBJ, FJT) between the two groups. Previous studies [59,60] involved participants of both sexes, aged between 6 and 18 years, and indicated an increase in SBJ performance in girls to a plateau at age 12–13 years. In fact, the study by Chung et al. [61], conducted on 12.712 Chinese students between the ages of 8 and 18 years, generated standard SBJ test data and showed stability of SBJ performance for female players from the age of 12 years old, while these values continue to increase for males up to 18 years with a very significant difference between the two sexes. Similarly, Ramírez-Vélez et al. [62] found that SBJ scores increased from the age of 9 to 12.9 years and reached a plateau at an age between 13 and 15.9 years for girls.

In another study, Ramos-Sepúlveda et al. [63] found further, better performances for males and which depended on muscle strength. We presume, therefore, that in the growth phase, the distinction according to sex is necessary to improve the explosive strength of the lower limbs, because this age interval is characterized by the linear increase in muscular explosive strength for both sexes. This latter is related to mechanical and neurological factors as previously shown [64]. Moreover, male subjects possess a more rapid increase in gonadal steroid hormones, growth hormones, and muscle mass and bone mineral content favoring better muscular strength performance until late adolescence [61]. In contrast, girls have an intense accumulation of fat mass in the early pubertal phase [16], which presents an inhibiting factor for muscle strength.

Concerning the agility *t*-test in U17 and U14 female handball players, the results showed better performance for U17 players with a highly significant difference compared to U14 players (*p* < 0.01 with a difference of 9%). The morphological factors are supposed to be related to maturation, such as height, the amount of fat mass, and muscle tissue mass [60].

Furthermore, the increases noted in the agility performance between U14 and U17 may have occurred in relation to the intermittent nature of the practiced sport involving multidirectional changes. Thus, regular practice in such a sport optimizes the quality of agility that naturally improves throughout childhood and adolescence, albeit in a non-linear fashion [65,66].

Lastly, the different measures of maximum aerobic speed (MAS) showed significant results that were better in U17 players compared to U14. This result may explain the importance of regular and continuous physical activities practiced during the preparation period, leading to improved aerobic capacity as indicated by other authors [67,68]. However, other studies reported conflicting results. For example, Berthoin et al. [69] showed that the MAS increased from 6 to 11 years for girls and then remained constant until 17 years for both sexes. Therefore, it is essential to take into account certain particularities such as sex or the intervention of certain biological processes (i.e., testosterone, GH, etc.) with a better evaluation of the aerobic capacity during the maturation phase.

However, the present study also has limitations. First, the number of participants studied is relatively small, and some subjects refused to join the study at the beginning, which may have led to selection bias in the results of the study. Second, perhaps the presence of a third group older than those studied could have added other interpretations. This will be taken into consideration in the following study. Third, only girls were included in this study, which didn’t allow us to better examine gender differences and show the effect of sleep on exercise performances. 

## 5. Conclusions

In conclusion, our collected results suggest the need to focus several factors, to assess physical abilities. It is therefore essential to employ a program in the field of sports practice adapted to young players, taking into account the biological changes related to their muscular development, while targeting qualities such as balance and strength, which are less developed, especially between 13 and 17 years. However, other studies seem necessary to examine the impact of the prevalence of sleep disorders in a larger and more varied population in order to improve the quality of games and ensure better performances.

## Figures and Tables

**Figure 1 ijerph-20-00330-f001:**
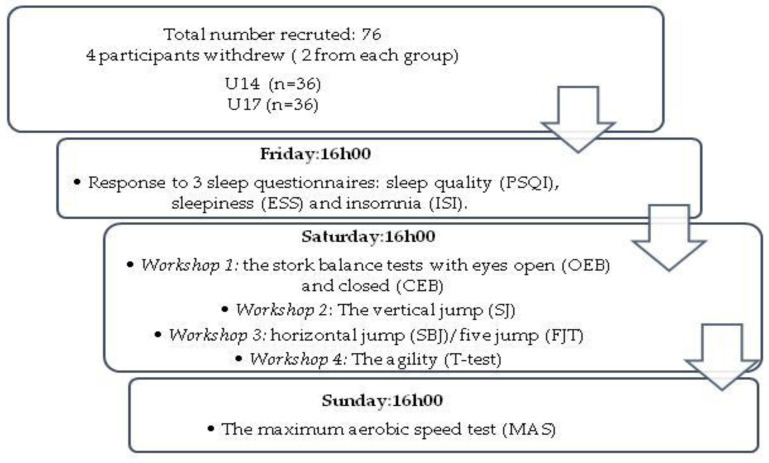
Study design for U14 and U17 female players.

**Table 1 ijerph-20-00330-t001:** Anthropometric parameters of the 2 groups of U14 and U17.

Group	U14 N = 36	U17 N = 36
PS (II–III/IV–V) Height (m)	15/21 1.64 ± 0.04	17/19 1.67 ± 0.05 *
Body mass (kg)	57.83 ± 5.79	59.21 ± 6.16 *
Body mass index (kg m^−2^)	21.49 ± 1.52	21.22 ± 1.63 *
Body fat (%)	24.45 ± 1.5	22.27 ± 1.3
Lean body mass (kg)	43.95 ± 2.7	46.18 ± 3.1 *

Values are mean ± SD; PS: pubertal stage. Significantly different from U14: * *p* < 0.05.

**Table 2 ijerph-20-00330-t002:** Comparison of the means of sleep quality, sleepiness, and insomnia between the two groups.

Group	PSQI	ISI	ESS
Group U14	5.44 ± 1.576	7.19 ± 2.649	6.36 ± 1.91
Group U17	5.47 ± 1.540	7.50 ± 2.49	6.31 ± 1.89

Values are mean ± SD; PSQI: Pittsburgh Sleep Quality Index; ISI: Insomnia Severity Index; ESS: Epworth Sleepiness Scale.

**Table 3 ijerph-20-00330-t003:** Percentages of sleep parameters of the U14 group and U17 group.

Variables	Group U14	Group U17	Average of the 2 Groups
PSQI ≥ 5	61.1%	63.8%	62.45%
PSQI < 5	38.9%	36.2%	37.55%
ISI ≥ 11	8.3%	8.4%	8.35%
ISI < 11	91.7%	91.6%	91.65%
ESS > 8	19.4%	22.3%	20.85%
ESS < 8	80.6%	77.7%	79.15%

PSQI ≥ 5: poor sleep quality; ISI ≥ 11: mild insomnia; ESS > 8: sleep debt or excessive daytime sleepiness.

**Table 4 ijerph-20-00330-t004:** Performance parameters of balance, agility, strength, and MAS in the two groups.

Group	OEB	CEB	SJ	SBJ	FJT	*t*-Test	VAMEVAL
U14	11.01 ± 8.52	2.88 ± 0.86	24.8 ± 5.38	1.59 ± 0.20	9.12 ± 0.54	12.8± 1.11	12.2 ± 1.35
U17	10.49 ± 9.75	2.24 ± 1.11	28.5 ± 4.04	1.66 ± 0.20	8.90 ±0.98	11.7 ** ± 0.59	12.69 * ± 0.91

Values are mean ± SD; OEB: open-eye balance; CEB: closed-eye balance; SJ: squat jump; SBJ: squat broad jump; FJT: five jump test; *t*-Test: Agility *t*-Test; VAMEVAL: maximum aerobic capacity test. Significantly different from U14: * *p* < 0.05, ** *p* < 0.01.

## Data Availability

In this study, the data presented are available on request from the corresponding author.

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
