# Peer review of "Evaluation of Age Based-Sleep Quality and Fitness in Adolescent Female Handball Players"

_ijerph, 2022, doi:10.3390/ijerph20010330_

Round 1

Reviewer 1 Report

The article includes good scientific information. However, there are some major concerns about the interest of this manuscript. 
- The title is not informative and does not reflect what is written in the introduction.

- No straightforward research question to be answered by the study. No justification for the aim of the study. The hypothesis is not clear. 

Most of the information mentioned in the Introduction does not direct the reader to the aim of the study. 

The methods were written with good details, but I was missing an exemplary flow chart of the study or graphs about some of the field tests performed to help the reader understand the outcome measure used. 

Reviewer 2 Report

Thank you for letting me review this intersting manuscript about the sleep hygiene and physical fitness in young female handball athletes. 

In general, I think is a well-written and well-described paper. The topic is interesting and the variables registered are okey. However, in my opinion, is little bit simple, could be improve including a larger sample size, more subgrups or comparing to other sports. 

Minor comments: 

- Line 103. Overall, the procedure and dependent variables are well-described. But, should be divided in two separate parts: the outcome variables and the procedure. Right now this part is miexd and is a little difficult to read and understand. 

- Line 148. Please add the name of the author

- Line 205. Only the t-test was used? all the variables were distributed normally? In case other statistical test were used, please write it down in the statistical analysis paragraph and in the tables of the results. 

- In general, the authors should revise all the abbreviatures of the manuscript. 

Reviewer 3 Report

Dear authors,

please check comments.

Kind regards

Round 2

Reviewer 1 Report

Thanks to the authors for making these changes. 

The introduction now includes justification for the study, but the length of the introduction is a bit long. Thus, I suggest reducing the length of the intro. 

Additionally, writing by (ref) was noticed a couple of times. Please cite the paper mentioned correctly without ignoring the names of the authors. 

The flow chart needs to include how many people were invited and how many participated. 

Please write in the footnote of the tables whether the presented data are i mean ± Standard deviation.

The discussion part. The first paragraph requires brief info on the main results found, which answers the research question raised at the start.

Author Response

Answers to the reviewer comments

Reviewer 1 :

  • Are all the cited references relevant to the research? Not applicable

As requested by the reviewer, we checked all the references and we found in the discussion section (line 303) one wrong reference: Han & Yang (2015), which was eliminated from the manuscript and from the list of references. We hope it is better now.

2- The introduction now includes justification for the study, but the length of the introduction is a bit long. Thus, I suggest reducing the length of the intro.

Done as suggested, and we reduced the length of the introduction. This gave more clarity. Thanks to the reviewer.

Some other changes were highlighted in yellow.

This has been deleted:

“…to determine the success of the performance in handball, it was necessary to resort to an exhaustive evaluation of these physical qualities in order to know the level of the players and to optimize the training of these qualities »

This has been deleted:

 “In this regard, authors showed that most physical measures are likely to be better in elite players compared to their sub-elite counterparts”

This was deleted “of adolescent males aged 14 to 19 years found that reactive agility quality may be a distinguishing factor for experienced versus amateur players »

and this was added « It was demonstrated however that the reactive agility quality presented a distinguishing factor for experienced versus amateur players. »

this was added « Although,  basic motor abilities were evaluated in women's handball (Srhoj et al., 2006) [13],  to the best of the authors' knowledge, no studies have examined the difference that may occur in some measures of the physical condition in the young players. »

in the place of : « In his study,(Srhoj et al., 2006) [13] evaluated the basic motor abilities that promote better decisive performance in women's handball, and suggesting fine coordination of the upper limbs as a main indicator of it.At this stage, and to our knowledge, there have been no studies examining the difference in some measures of physical condition between 2 groups of female players of different ages in the maturation phase.”

This was deleted: “Through our study, we wondered if the specificity of sex in the phase of puberty can lead us to new alterations of the physical condition between U14 and U17.”

This was further deleted: “Furthermore, in addition to the evaluation of physical parameters, it is necessary to evaluate external parameters such as sleep hygiene which represents one of the key factors of performance.”

3- Additionally, writing by (ref) was noticed a couple of times. Please cite the paper mentioned correctly without ignoring the names of the authors.

Thanks again to the reviewer; this has been done as suggested. We think that this is better in the manuscript and is in line with the journal demands like cited in the “Instructions to Authors.”

4- The flow chart needs to include how many people were invited and how many participated.

This was added as suggested in the flow chart. This was further added in the methods section and was highlighted in yellow. “It should be noted that four participants withdrew before the start of the study for personal reasons (2 from each group) and their data were not accounted for in the statistical analysis.”

We hope it is better now.

5-Please write in the footnote of the tables whether the presented data are i mean ± Standard deviation.

Done as suggested. This was added: Values are mean ± SD

6- The discussion part. The first paragraph requires brief info on the main results found, which answers the research question raised at the start.

Done as suggested. Thanks to the reviewer, we think that this has improved the discussion section and give an inf on the main results.

Reviewer 3 Report

Authors have significantly improved their manuscript. I am satisfied with most changes and I proceed to the Editor with the decision.

Kind regards

Author Response

The authors want to thank  the reviewer for his participation in increasing the quality of the manuscript with his valuable comments and for being satisfied with the modifications related to the text.
